# Enhanced Tensile Strength of Monolithic Epoxy with Highly Dispersed TiO₂-Graphene Nanocomposites

**Yanshuai Wang** [1,*] **, Siyao Guo** [2]**, Biqin Dong** [1] **and Feng Xing** [1]

1 Guangdong Province Key Laboratory of Durability for Marine Civil Engineering,
College of Civil and Transportation Engineering, Shenzhen University, Shenzhen 518060, China;
incise@szu.edu.cn (B.D.); xingf@szu.edu.cn (F.X.)

2 Cooperative Innovation Center of Engineering Construction and Safety in Shandong Blue Economic Zone,
School of Civil Engineering, Qingdao University of Technology, Qingdao 266033, China; siyaoguo@126.com

* Correspondence: yswang@szu.edu.cn; Tel.: +86-755-8696-9524

**Abstract:** The functionalization of graphene has been reported widely, showing special physical and chemical properties. However, due to the lack of surface functional groups, the poor dispersibility of graphene in solvents strongly limits its engineering applications. This paper develops a novel green "in-situ titania intercalation" method to prepare a highly dispersed graphene, which is enabled by the generation of the titania precursor between the layer of graphene at room temperature to yield titania-graphene nanocomposites (TiO₂-RGO). The precursor of titania will produce amounts of nano titania between the graphene interlayers, which can effectively resist the interfacial van der Waals force of the interlamination in graphene for improved dispersion state. Such highly dispersed TiO₂-RGO nanocomposites were used to modify epoxy resin. Surprisingly, significant enhancement of the mechanical performance of epoxy resin was observed when incorporating the titania-graphene nanocomposites, especially the improvements in tensile strength and elongation at break, with 75.54% and 176.61% increases at optimal usage compared to the pure epoxy, respectively. The approach presented herein is easy and economical for industry production, which can be potentially applied to the research of high mechanical property graphene/epoxy composite system.

**Keywords:** graphene; in-situ titania intercalation; dispersity; epoxy; tensile strength





## 1. Introduction

Graphene is considered one of the most powerful materials in the 21st century because of its unique sp2 and honeycomb structures [1]. Such molecular structures empower the graphene materials with excellent mechanical [2], thermal [3], and electrical properties [3] for maximum utilization in various fields, e.g., as transparent conductive films [4,5], sensors [5,6], and clean energy devices [5,7]. Reduced graphene oxide (RGO) is a form of acquisition of graphene which can be obtained from the chemical reduction in graphene oxide (GO) [1]. The RGOs with improved and diverse performances can be produced when the GOs are subjected to various reductors and surface modifiers. Despite these superiorities, the application of graphene is still limited. Due to the strong Van der Waals' force [8] from the interlayers of graphene, nano-sized graphene is prone to agglomerate with each other. Therefore, dispersion of graphene was always a vital problem in recent studies. In order to achieve well-dispersed and highly-performed graphene for optimal enhancement in the properties of graphene composites, numerous improvements have been conducted in recent years. Generally, using surfactants [9–12] and solvents [13] are two effective methods for dispersing graphene. The surfactants are expected to modify the graphene by incorporating functional groups or polymers onto the surface of graphene. For example, the sodium cholate as a surfactant enabled the production of the high-quality graphene with a stable and easy film [10]. Additionally, the poly(sodium 4-styrenesulfonate) (PSS) can be chemically blended with the graphene for good dispersion through surface treatment of

the graphene. The PSS-coated graphene exhibited stable and high-efficiency dispersion in any solvent authors used [11]. Some functional groups, nanoparticles and polymers, were employed to react with the graphene to obtain the well-dispersed graphene or graphene derivatives. By introducing SO3- groups onto the surface of graphene, a water-soluble sulfonated graphene was produced, which showed higher electrical conductivity compared to graphite [14]. A facile solution casting method has been used to obtain reduced graphene oxide/alumina (RGO/Al) with better dispersion in carboxymethyl cellulose (CMC) to produce RGO-Al-CMC composites. Such an action significantly improves the tensile strength and toughness of RGO-Al-CMC composites (about 4.3–6.7 times upscaling compared to the control matrix [15]. The employment of an efficient solvent in improving graphene dispersion also has good effects. A liquid-phase exfoliation for the graphene on the basis of compound solvent of ethanol and water has been exploited. No destruction and good dispersibility exhibited in the prepared graphene that showed prospects in the electronic, energy and optical fields [16]. However, the toxicity of the used solvent and surfactant may impact side influences on the performance of the graphene materials [17]. It is imperative to develop other methods for the graphene dispersion without redundant and harmful surfactants.

The epoxy resin (EP) as a common coating material has been extensively applied all over the world. Excellent mechanical properties, adhesion and corrosion resistance largely popularized epoxy resin in many fields, such as civil engineering [18], aerospace [19] and electronic materials [20]. However, the high crosslink of hardened epoxy resin leads to the brittleness and low toughness, which have become major problems that restrict the development and application of the epoxy resin. In recent years, scholars have made a lot of efforts to improve the performance of epoxy resin. Incorporating fibers and nanoparticles to improve the toughness of epoxy resin has led to some achievements. Natural fibers (bamboo fiber [21,22], banana fiber [23] and cellulose fiber [24]), carbon fibers [25–27], glass fiber [28], nanoclays [29], carbon nanotubes [30] and nano metal oxides [31] can improve toughness, thermal stability and other performance of epoxy. However, the poor expanding and flexural properties are still the main drawbacks of epoxy resin to be solved presently. It has been reported that graphene can enhance the mechanical performance of epoxy resin [32,33].

In this work, we propose a green and one-step way to obtain the well-dispersed $TiO_2$-RGO nanocomposites, which were further applied in the epoxy resin for the improvement of its mechanical properties. The transmission electron microscopy (TEM), Fourier transform infrared (FTIR) spectra and Raman spectra have been utilized to investigate the morphologies of the highly dispersed $TiO_2$-RGO nanocomposites and the titania-graphene-epoxy ($TiO_2$-RGO-EP) composites. The tensile strength and elongation at break of $TiO_2$-RGO-EP composites were measured to verify the efficacy of the graphene on the mechanical performance of the epoxy resin. The result shows that the $TiO_2$-RGO nanocomposites can bridge the cracks during the epoxy curing and tradeoff the tensile impact of the epoxy; the strengthening mechanism of $TiO_2$-RGO nanocomposites on the epoxy resin has been deeply discussed.

## 2. Materials and Methods

### 2.1. Raw Materials

In this work, the raw materials used for producing GO were natural graphite flakes (325 mesh, 99.95% pure), which were purchased from Laixi Carbon Sources Trade Department (Qingdao, China). Other reagents for the oxidation of graphite flakes, such as the concentrated $H_2SO_4$ solution with a mass fraction of 98%, HCl solution (mass fraction of 35–36%), $NaNO_3$, $KMnO_4$ and $H_2$, and the solvent (i.e., dimethylbenzene and N-butanol), were all supplied by Sinopharm Chemical Reagent Co., Ltd. (Shanghai, China). The sodium borohydride (NaBH4), anatase titanium oxide ($TiO_2$) and ethanol were all purchased from Sinopharm Chemical Reagent Co., Ltd. (Shanghai, China). The epoxy was standard diglycidyl ether of bisphenol-A (type E-44), and polyamide resin-650 as the matched hardener

was used to prepare monolithic epoxy matrix. Both were bought from Jinan Tianmao Resin Chemical Co. Ltd. (Shandong, China).

## 2.2. Preparation of GO, RGO and TiO$_2$-RGO Nanocomposites

The graphene oxide (GO) which was used to synthesize the titania-graphene (TiO$_2$-RGO) nanocomposites was obtained by the modified hummers method [34], as shown in Figure 1. As formulated in Table 1, the flake graphite was oxidized in a solution of sulfuric acid and sodium nitrate at the temperature of 0 °C, following the slow addition of potassium permanganate. It should be noted that the mixed solution was stirred for complete oxidation with a low speed rate for 12 h. An amount of 90 mL of deionized water was added into the flask for continuous 20 min stirring at an environment temperature of 98 °C. A luminous yellow solution was exhibited when the residual oxidants were removed by hydrogen peroxide solution. The dried GO was obtained at the vacuum oven after repeated washing by the hydrochloric acid solution.

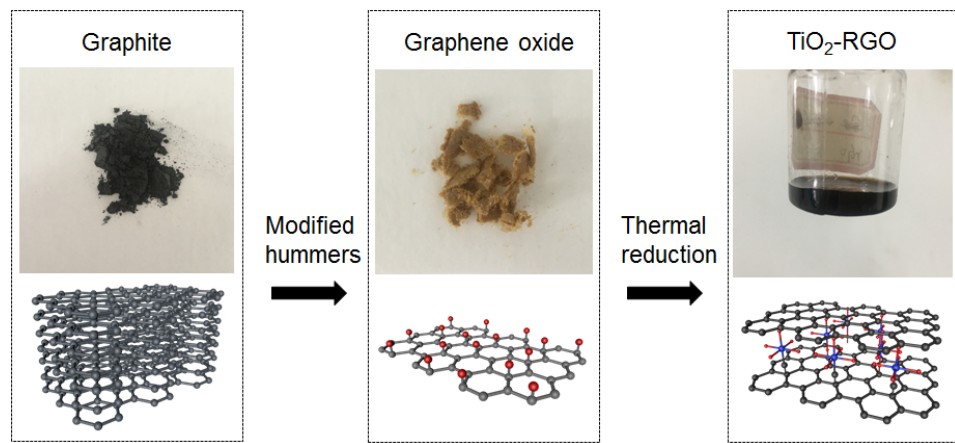

**Figure 1.** Schematic diagram for the synthesis process of TiO$_2$-RGO nanocomposites.

**Table 1.** Mixture proportion for the modified Hummers method.

| Raw Material | Flake Graphite | NaNO$_3$ | H$_2$SO$_4$ | KMnO$_4$ |
|:---:|:---:|:---:|:---:|:---:|
| dosage | 2.0 g | 1.5 g | 67.5 mL | 9.0 g |

The synthesized GO (0.1 g) was sonicated in the deionized water (100 mL) for 2 h until it became a well dispersed GO solution. The sodium borohydride (NaBH4)-dissolved ethanol solution (1 g NaBH4 in 100 mL ethanol) and the precursor of titanium oxide (10% of GO dosage, i.e., 0.01 g) were alternately added to the GO solution with a continuous magnetic stirring under a 65 °C water bath for 12 h. As a comparison, the same sodium borohydride (NaBH4)-dissolved ethanol solution was used for reduction graphene (RGO) preparation under the same reaction environment. The TiO$_2$-RGO nanocomposites and RGO particles were respectively collected after sequential filtering, washing, drying and grinding.

## 2.3. Fabrication of TiO$_2$-RGO-EP Composites

The schematic diagram for the preparation of titania-graphene-epoxy composites (TiO$_2$-RGO-EP) is shown in Figure 2. It was of critical importance to obtain a homogenous epoxy with TiO$_2$-RGO nanocomposites. A complex solvent with a xylene-to-N-butanol weight ratio of 7:3 was used to disperse the TiO$_2$-RGO nanocomposites and E-44 epoxy, respectively. The TiO$_2$-RGO solution was then mixed with the epoxy solution that was preheated at 80 °C for 2 h. Afterwards, the hardener was homogenized in TiO$_2$-RGO-EP solution with a weight ratio of 1:1 to fabricate the TiO$_2$-RGO-EP composites. The final mixture was cast into the molds for 1 h degassing and 3 h thermal curing in an oven

(65 °C). The hardened monolithic epoxy matrices were demolded for tensile testing and microanalysis. The detailed experimental program flow is displayed in Figure 3.

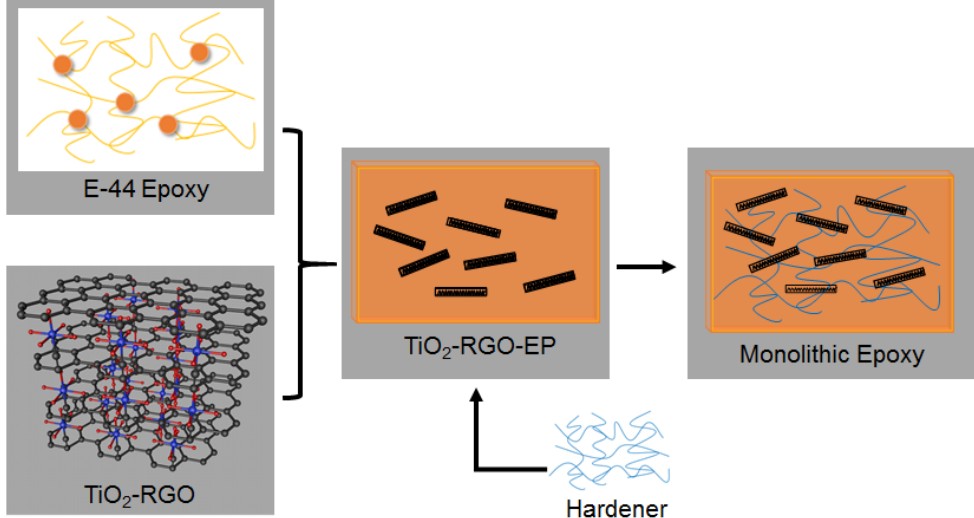

**Figure 2.** Schematic diagram for the preparation of TiO$_2$-RGO-EP composites.

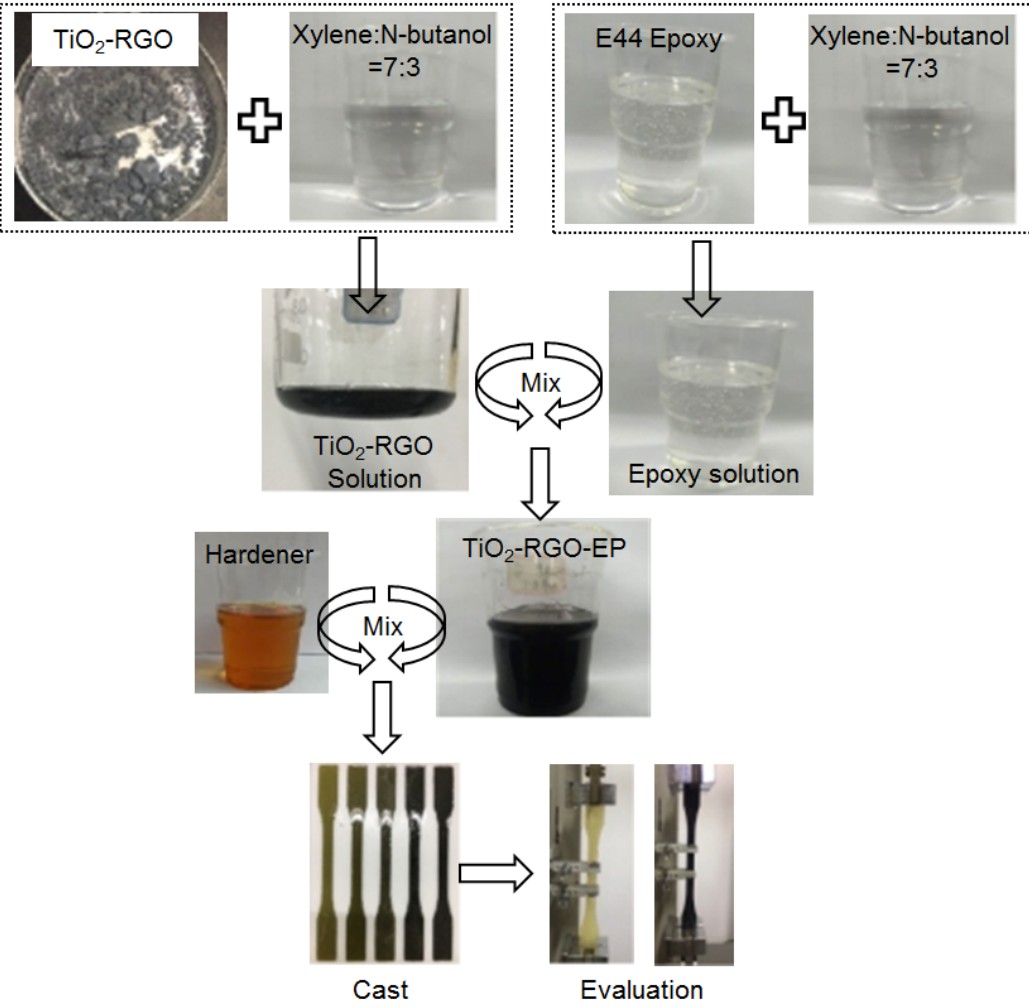

**Figure 3.** Experimental program flow.

The schematic diagram for the preparation of titania-graphene-epoxy composites ($TiO_2$-RGO-EP) is shown in the Figure 2. It was of critical importance to obtain a homogenous epoxy with $TiO_2$-RGO nanocomposites. A complex solvent with a xylene-to-N-butanol weight ratio of 7:3 was used to disperse the $TiO_2$-RGO nanocomposites and E-44 epoxy, respectively. The $TiO_2$-RGO solution was then mixed with the epoxy solution that was preheated at 80 °C for 2 h. Afterwards, the hardener was homogenized in $TiO_2$-RGO-EP solution with a weight ratio of 1:1 to fabricate the $TiO_2$-RGO-EP composites. The final mixture was cast into the molds for 1 h degassing and 3 h thermal curing in oven (65 °C). The hardened monolithic epoxy matrices were demolded for tensile testing and microanalysis. The detailed experimental program flow is displayed in Figure 3.

### 2.4. Material Characterization & Performance Assessment

The morphologies and disperse states of $TiO_2$-RGO solution and $TiO_2$-RGO-EP solution (xylene and N-butanol as solvent, as shown in Figure 3) were characterized by the transmission electron microscopy (TEM). Fourier transform infrared (FTIR, Thermo Fisher Scientific Nicolet) and Raman (Renishaw inVia 71J012) spectroscopies were conducted to investigate the molecular structures of $TiO_2$-RGO nanocomposites and $TiO_2$-RGO-EP composites. For the FTIR measurement, the samples were ground together with the KBr powder, and the mixed powder was pressed into a transparent sheet for test. FTIR spectroscopy results were collected by absorption mode from 4000 to 400 cm$^{-1}$ with a resolution of 4 cm$^{-1}$.

The tensile strength of $TiO_2$-RGO-EP composite matrices were tested by a computer controlled universal material testing machine (MZ-4000D) with a speed rate of 2 mm/min. The shape and dimension of samples for tensile test is designed according to GB/T2567-2008, as shown in Figure 4. The tensile strength and elongation of samples were calculated following Equations (1) and (2), respectively. The maximum tensile strength and elongation at break were both obtained during the test.

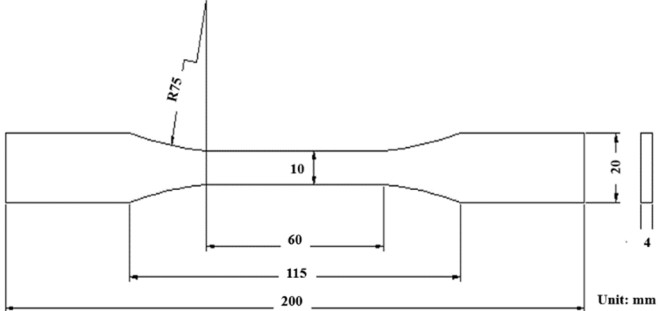

**Figure 4.** The shape and dimension of samples for tensile test.

The microstructures of $TiO_2$-RGO-EP composite matrices with different dosages of $TiO_2$-RGO nanocomposites were analyzed by scanning electron microscopy (SEM, ZEISS MERLIN) to explore the influences of $TiO_2$-RGO nanocomposites on final epoxy products. Before the microscopic measurement, the typical samples were desiccated (60 °C for 24 h) and surface coated with gold sputter.

$$Strength = \frac{Force_{max}}{Sectional\ area} \tag{1}$$

$$Elongation = \frac{\Delta L}{L} \times 100\% \tag{2}$$

## 3. Results and Discussion

### 3.1. Morphological Features of RGO, TiO$_2$-RGO and TiO$_2$-RGO-EP

As shown in Figure 4, the macro- and micro-morphological features and disperse states of RGO, TiO$_2$-RGO and TiO$_2$-RGO-EP are compared. Figure 5a shows the distribution change in RGO and TiO$_2$-RGO in the aqueous solution along with the standing time (0, 10 and 20 min). In the beginning, both solutions behave highly dispersed under supersonic vibration. The remarkable sinking was observed in the RGO solution after 10 min standing, following a total separation with dual zones (i.e., RGO and water) in 30 min. In contrast, the TiO$_2$-RGO solution remain unchanged during the same period, which meant the introduction of TiO$_2$ nanoparticles enabled a well disperse state of RGO solution. It is presumed that the graphene sheets are well embedded between TiO$_2$ nanoparticles to prevent the great van der Waals forces of RGO particles. The structure directing features of the incorporated layered graphene component may cause the TiO$_2$ nanoparticles with marginal usage in TiO$_2$-RGO nanocomposites (i.e., 10% of GO dosage) to be perceived. Figure 5b shows the TEM image of the highly dispersed graphene. It can be seen that the highly disperse graphene is so thin that it looks similar to a cicada's wing. The translucent two-dimensional tiled state graphene sheets are stacked on each other with a pleated appearance. Figure 5c shows the nano-morphologies of TiO$_2$-RGO-EP composites. The diluted epoxy microscopically presents a spherical shape, as marked in Figure 5c. It can also be seen that TiO$_2$-RGO nanoparticles as the "nanofillers" were combined with the epoxy that changed the distribution pattern of epoxy. The RGO no longer existed in a flat form, and the epoxy complexed RGO underwent a certain degree of twisting, folding and bending, as shown in TEM image.

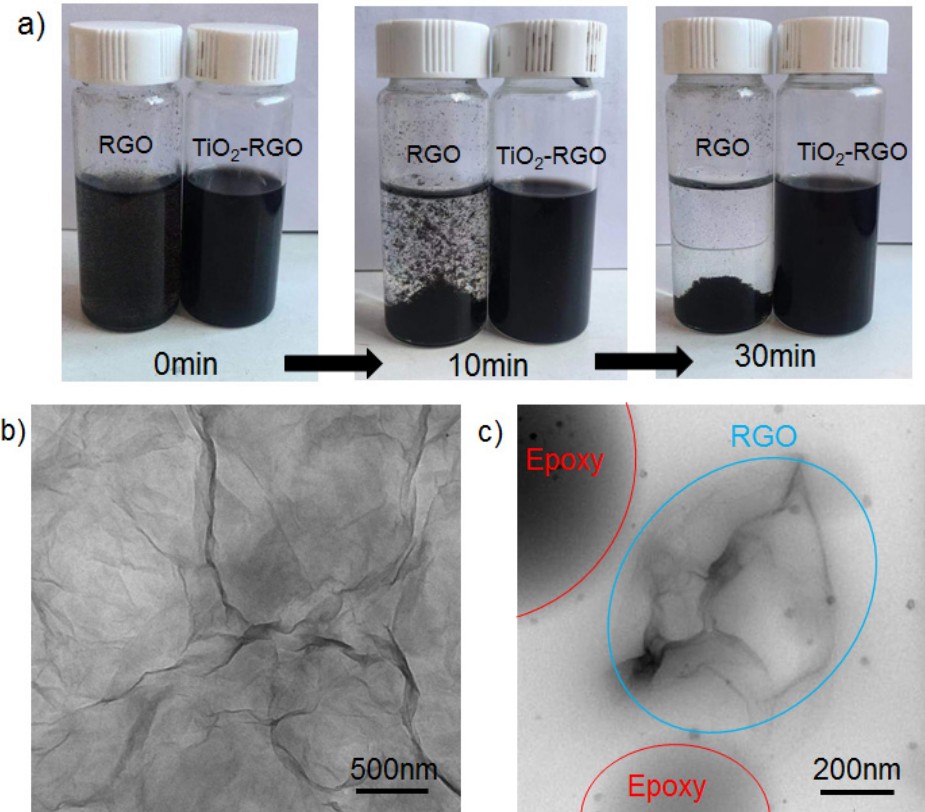

**Figure 5.** The macro- and micro-morphologies and disperse states of RGO, TiO$_2$-RGO and TiO$_2$-RGO-EP. (**a**) Macroscopic observation of disperse state; (**b**) TEM image TiO$_2$-RGO nanoparticles; (**c**) TEM image of TiO$_2$-RGO-EP composites.

### 3.2. FT-IR and Raman Analysis of TiO$_2$-RGO Nanoparticles

Structural characterization of the TiO$_2$ and TiO$_2$-RGO nanoparticles were explained by FTIR and Raman spectroscopy. As indicated in Figure 6a, the TiO$_2$ well exhibited the characteristic absorption peaks at 3420, 1563 and 1451 cm$^{-1}$ and continuous peaks from 800 to 500 cm$^{-1}$, which correspond to the absorption peaks of TiO$_2$. Moreover, the intensity of the defined absorption peaks of 3420 cm$^{-1}$ on TiO$_2$-RGO nanoparticles are a little weaker than those of TiO$_2$, and the intensity of the peak at 700 cm$^{-1}$ and 400 cm$^{-1}$ obviously increased. The above changes in absorption peak intensity can be attributed to the synthesis of Ti-O bonds and T-C bonds [35]. Additionally, the transformations of intensity and position of defined peaks indicates chemical bonding has been formed between TiO$_2$ and RGO.

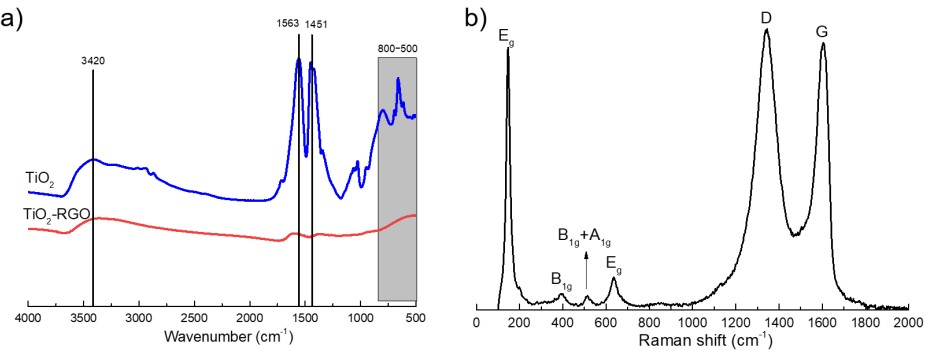

**Figure 6.** FTIR (**a**) and Raman spectra (**b**) of the TiO$_2$-RGO nanoparticles.

The Raman spectra of the TiO$_2$-RGO samples has been shown in Figure 6. The characteristic band of carbon material is the D band, used to characterize the structural defects or edges in graphene samples, which always shows up at about 1350 cm$^{-1}$, and the G band, which always shows up at about 1580 cm$^{-1}$, represents the in-plane vibration of sp2 carbon atoms [2,36]. It can be seen from the Figure 6b that the D band of the TiO$_2$/RGO is shown at 1339 cm$^{-1}$, and G band of the same is 1602 cm$^{-1}$. Besides the D band and G band, the nanocomposites also have four peaks appearing at 146, 389, 514 and 636 cm$^{-1}$, which are characteristic of the e.g., B1g(1), A1g and B1g(2), and Eg(2) modes of anatase TiO$_2$, respectively [37]. Combined with the results of the FT-IR, it is known that titanium dioxide is successfully intercalated between the layers of graphene sheets.

In order to identify the combination between TiO$_2$-RGO and EP, the Fourier transform infrared spectroscopy (FTIR) and Raman spectroscopies were utilized to study the functional groups and molecular skeleton of TiO$_2$-RGO-EP composites. FTIR of the pure EP and TiO$_2$-RGO nanocomposites-modified EP (0.10% addition) is shown in Figure 7a. The spectrum of the resulting TiO$_2$-RGO-EP composites showed C-H stretching vibration at 3055 cm$^{-1}$ from the benzene ring in the skeleton of EP. The EP also shows absorption peaks at 2965, 2927 and 1456 cm$^{-1}$, indicating the asymmetric shrinkage peak of the methyl group, asymmetric contraction peak of the methylene group and asymmetrical bending of methyl. The C-C stretching vibration is located at 1607, 1581 and 1509 cm$^{-1}$. There are also C-(CH$_3$)$_3$ skeleton vibration, C-H$_2$ rocking vibration, C-H swing vibration of C-H and in-plane deformation of (CH$_2$)$_3$ occurring at 1246, 915, 815, 772 and 736 cm$^{-1}$, respectively. In addition to the skeleton structure, the characteristic peak of EP was detected at 1036 cm$^{-1}$, which indicated the C-O-C vibration [3]. In conclusion, the marginal change in peaks of FTIR of TiO$_2$-RGO-EP composites was identified compared to the pure EP, which meant that no chemical bonding was produced due to the addition of TiO$_2$-RGO nanocomposites.

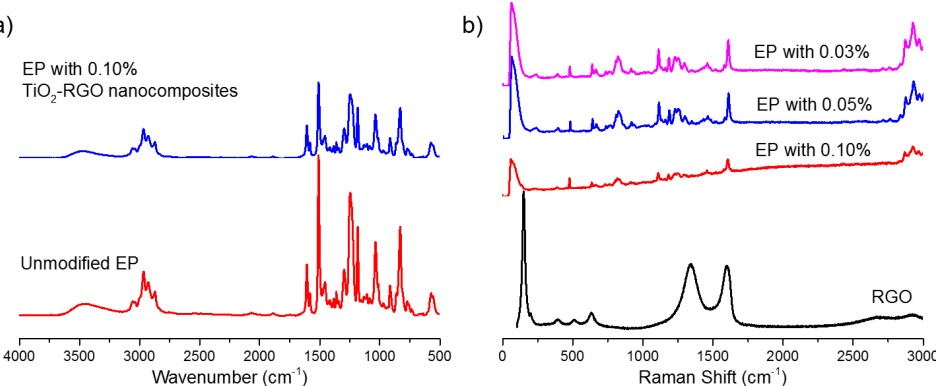

**Figure 7.** FTIR (**a**) and Raman spectra (**b**) of unmodified EP and TiO$_2$-RGO-EP composites.

Raman spectra of the TiO$_2$-RGO-EP composites with different percentages of TiO$_2$-RGO nanocomposites are shown in Figure 7b, where the Raman spectrum of the prepared TiO$_2$-RGO nanocomposites is inserted as a reference. Different from the unmodified EP sample, the Raman spectra of epoxy resin samples with the TiO$_2$-RGO nanocomposites all exhibit two characteristic bands caused by these nanomaterials. It was found that the frequency of the G band shifted from 1602 to 1581 cm$^{-1}$. At the same time, the frequency of D band shifted from 1339 to 1346 cm$^{-1}$. The appearance of the D and G bands both resulted from the doping of TiO$_2$-RGO nanocomposites.

### 3.3. Mechanical and Morphological Properties of TiO$_2$-RGO-EP Composites

Tensile strength and elongation are two intuitive indicators that confirm the effect of the TiO$_2$-RGO nanocomposites on the mechanical improvement of the epoxy matrix. The test results of the TiO$_2$-RGO-EP composites varying with the dosage of TiO$_2$-RGO nanocomposites are shown in Figure 8. As shown in Figure 8a, the tensile strength of the TiO$_2$-RGO-EP composites achieved the maximum increase (75.54% increase compared to the control one and reaching 36.09 MPa) in case of 0.10% TiO$_2$-RGO nanocomposites. Other groups with the dosages of such nanocomposites (i.e., 0.03%, 0.05% and 0.20%) could obtain increases in tensile strength to 27.62, 32.03, and 32.09 MPa, respectively, compared to the epoxy without additives (20.56 MPa). When the content of graphene in epoxy exceeded the optimum dosage (i.e., 0.10% TiO$_2$-RGO nanocomposites), the tensile strength of the final products decreased, which meant that the superfluous graphene would compromise the performance of monolithic epoxy.

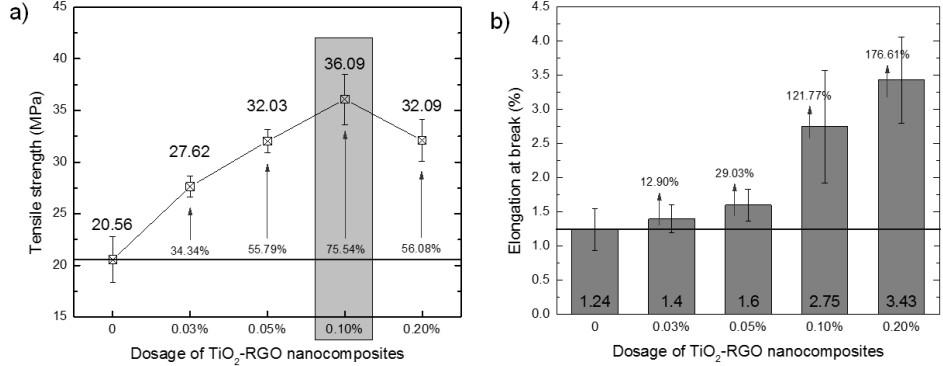

**Figure 8.** Tensile strength (**a**) and elongation at break (**b**) of TiO$_2$-RGO-EP composites.

The elongation at break indicates the toughness of the epoxy resin. The elongation of the TiO$_2$-RGO-EP composites varying with the dosage of TiO$_2$-RGO nanocomposites is shown in Figure 8b. The control epoxy (i.e., without any TiO$_2$-RGO nanocomposite) has a 1.24% elongation at break. The elongation of the TiO$_2$-RGO-EP composites improve

significantly with the increase in $TiO_2$-RGO nanocomposites. The maximum elongation at break was achieved when the dosage of nanocomposites was 0.2%, reaching 3.43%, which was 176.6% higher than that of the control epoxy. The elongation of the $TiO_2$-RGO-EP composites with 0.03% and 0.05% RGO did not show impressive increase (only 12.9% and 29.0%, respectively).

SEM images of the $TiO_2$-RGO-EP composites with different dosages of $TiO_2$-RGO nanocomposites are displayed in Figure 9. It was clearly found that the morphologies of break section of epoxy resin samples were greatly changed with the addition of $TiO_2$-RGO nanocomposites. In the control epoxy, the break section had a regular texture and was smooth at high magnification without excess protrusion and wrinkles. After amplification, micro cracks can be seen in the tensile section. On the contrary, the graphene-modified epoxy exhibited an irregular texture and a rough section at the break sections, regardless of graphene dosage.

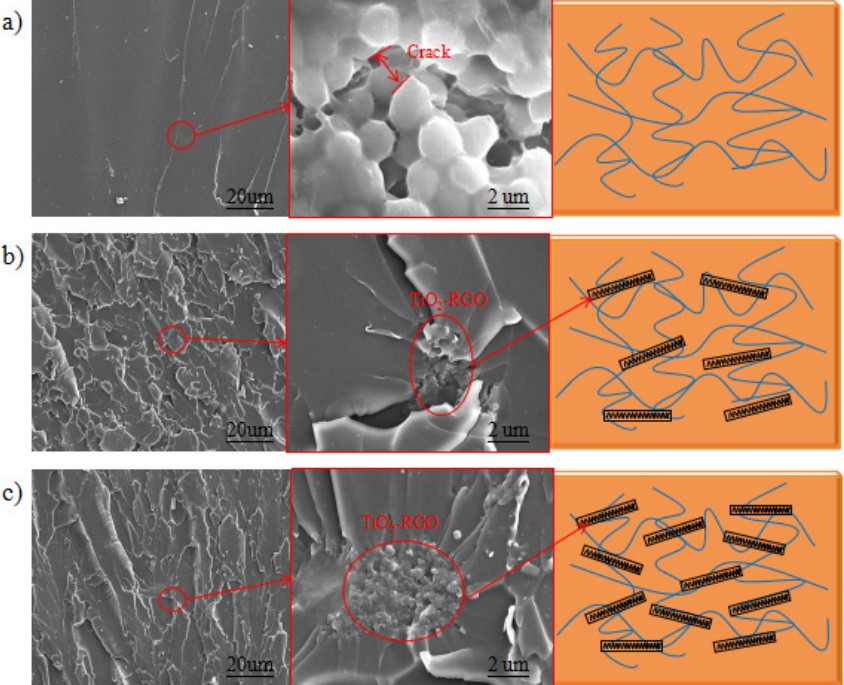

**Figure 9.** SEM images of the break section of $TiO_2$-RGO-EP composites with different dosage of $TiO_2$-RGO nanocomposites. (**a**) 0%, (**b**) 0.03%, (**c**) 0.10%.

The seeding of anatase $TiO_2$ nanoparticles into RGO produced the $TiO_2$-RGO nanocomposites with highly improved dispersion of RGO. Such chemical synthesis enabled the anatase $TiO_2$ to insert the interlayers of the RGO. The dispersion of nano-sized $TiO_2$-RGO materials in the epoxy bridged the cracks generated during the epoxy curing, which may lead to the change in the transmission path of the epoxy sample under the concentrated load. The strengthening mechanism of $TiO_2$-RGO nanocomposites on the tensile test of epoxy is shown in Figure 10. The $TiO_2$-RGO nanocomposites can tradeoff the tensile effect in the tensile samples. The stretched sections with rough and uneven surface are left behind due to the pull-out RGO nanoparticles [38]. Thus, a significant increase in the tensile properties of the graphene-modified epoxy with optimal dosage resulted.

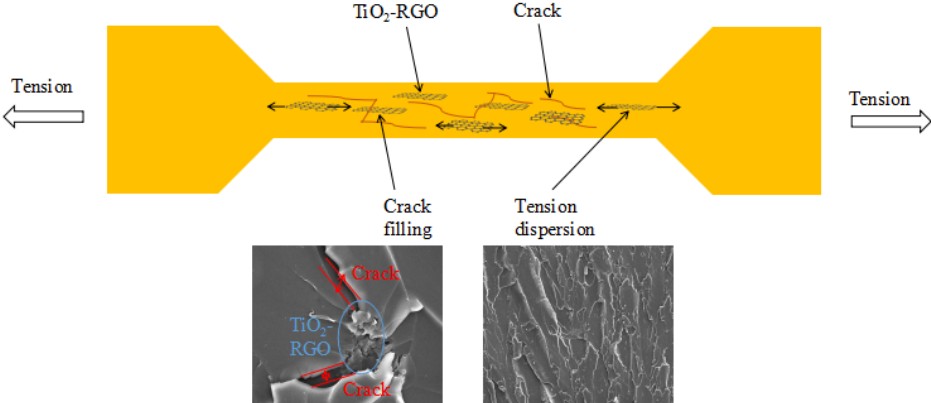

**Figure 10.** Strengthening mechanism of TiO$_2$-RGO nanocomposites on the tensile test of EP.

## 4. Conclusions

This paper reported the synthesis of a kind of highly dispersed graphene by using a novel "in-situ titania intercalation" method, which enabled inserting the nano titania between the layer of graphene to overcome the interfacial van der Waals force of graphene. These features were confirmed by the FTIR and Raman spectroscopies. Such highly disperse titania-graphene nanocomposites were further used to modify epoxy resin by a simple method. Compared to the unmodified epoxy, the optimal tensile strength and elongation of the TiO$_2$-RGO-EP composites can increase by 75.54% and 121.77% in case of a 0.10% addition of TiO$_2$-RGO nanocomposites. Additionally, the elongation of the epoxy could reach 176.61% when 0.20% TiO$_2$-RGO nanocomposites were used. The surprising result can solve the main drawbacks of epoxy resin (such as the poor expanding and flexural properties). This highly dispersed titania-graphene nanocomposites could be expected to be applicable in many other fields due to their merits, as they have a low cost, simple procedure and good mechanical property.

**Author Contributions:** Conceptualization, Y.W. and S.G.; methodology, Y.W.; formal analysis, S.G.; writing—original draft preparation, Y.W.; writing—review and editing, B.D. and F.X.; supervision, F.X.; funding acquisition, S.G. and F.X. All authors have read and agreed to the published version of the manuscript.

**Funding:** This research was funded by the National Natural Science Foundation of China (No. 51978354). The authors would like to acknowledge the financial support from Guangdong Provincial Key Laboratory of Durability for Marine Civil Engineering (No. 2020B1212060074) and from Guangdong Provincial Education Department Project (No. 2018KZDXM060).

**Acknowledgments:** The authors would like to acknowledge the financial support from the National Natural Science Foundation of China (No. 51978354), Guangdong Provincial Key Laboratory of Durability for Marine Civil Engineering (No. 2020B1212060074), and Guangdong Provincial Education Department Project (No. 2018KZDXM060).

**Conflicts of Interest:** The authors declare no conflict of interest.

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
