# Peer review of "Enhanced Tensile Strength of Monolithic Epoxy with Highly Dispersed TiO2-Graphene Nanocomposites"

_jcs, doi:10.3390/jcs5070191_

Round 1

Reviewer 1 Report

In this manuscript, Wang et al. have reported the synthesis and preparation of epoxy with TiO2-graphene fillers and investigated the mechanical properties of the composites. The paper has gone through a thorough characterization of such composites. The English of the manuscript should be checked by a native speaker. For example, there are some incomplete sentences (lines 245, 246) and other minor grammatical issues. Apart from that, authors should address the following questions:

  • The caption for Figure 8 is missing.
  • I can see that the maximum tensile strength is achieved at 0.10% and then is decreased in the 0.20%. What is the reason for that? Is it because of synergistic effects? What is the ratio between TiO2 and graphene fillers, i.e. what percentage of the filler is TiO2 and what percentage is graphene?
  • The authors also should explain how the error bars shown in Figure 8 are calculated.
  • Figure 4 can be moved to the Supplementary Information. Its presence in the main text is redundant.
  • In TEMs shown in Figure 5, where are the TiO2 nanoparticles?
  • What do Raman measurements exactly prove in this manuscript? The defects are induced to graphene flakes as soon as they are incorporated in any matrix which causes D-band to appear. I cannot see any clear conclusions from Raman experiments.
  • How does Raman or FTIR can prove the combination of TiO2-rGo? I can understand that the authors have observed the Raman peaks of each constituent material but could not understand how they used these two techniques to prove the attachment/combination of these two fillers.

Overall, the manuscript is fit to be published after the above comments are addressed in full.

Author Response

In this manuscript, Wang et al. have reported the synthesis and preparation of epoxy with TiO2-graphene fillers and investigated the mechanical properties of the composites. The paper has gone through a thorough characterization of such composites. The English of the manuscript should be checked by a native speaker. For example, there are some incomplete sentences (lines 245, 246) and other minor grammatical issues. Apart from that, authors should address the following questions:

Response: Thanks for your examination. The language of the manuscript is refined.

  • The caption for Figure 8 is missing.

Response: Thanks for your examination. The caption for Figure 8 is added.

  • I can see that the maximum tensile strength is achieved at 0.10% and then is decreased in the 0.20%. What is the reason for that? Is it because of synergistic effects? What is the ratio between TiO2 and graphene fillers, i.e. what percentage of the filler is TiO2 and what percentage is graphene?

Response: Thanks for your examination. The caption for Figure 8 is added. The effect of TiO2 in TiO2-RGO nanocomposites aims to increase the degree of dispersion of RGO, which is used for improve the tensile strength of monolithic epoxy. For the epoxy with 0.2% dosage of nanocomposites, its tensile strength is decreased, because more dosage influences the cross-linked effects of epoxy in the results of the drop of mechanical strength.

  • The authors also should explain how the error bars shown in Figure 8 are calculated.

Response: Thanks for your examination. The final strength is obtained from the average value of three measurements, and the error bar is the difference value of maximum value and minimum value.

  • Figure 4 can be moved to the Supplementary Information. Its presence in the main text is redundant.

Response: Thanks for your advice. I think the figure 4 will provide a clear geometry of the tested samples for tensile strength. I prefer to keep it in the main text.

  • In TEMs shown in Figure 5, where are the TiO2 nanoparticles?

Response: The TiO2 nanoparticles are inserted into the layer of graphene, and are invisible in TEM.

  • What do Raman measurements exactly prove in this manuscript? The defects are induced to graphene flakes as soon as they are incorporated in any matrix which causes D-band to appear. I cannot see any clear conclusions from Raman experiments.

Response: The results of Raman measurements aim to prove that the insertion of TiO2 in RGO (i.e., TiO2-RGO nanocomposites) does not influence the characteristic absorption peaks of RGO.

  • How does Raman or FTIR can prove the combination of TiO2-rGo? I can understand that the authors have observed the Raman peaks of each constituent material but could not understand how they used these two techniques to prove the attachment/combination of these two fillers.

Response: The results of Raman and FTIR both aim to prove that the insertion of TiO2 in RGO (i.e., TiO2-RGO nanocomposites) does not influence the characteristic features of RGO. The attachment/combination of TiO2 is only for good dispersion of RGO instead of physical or chemical features of RGO.

Overall, the manuscript is fit to be published after the above comments are addressed in full.

Reviewer 2 Report

Well writen and interesting topic. The intended application is missing.

Author Response

Well writen and interesting topic. The intended application is missing.

Response: Thanks for your advice. The good dispersion of RGO is achieved through the insertion of TiO2. The epoxy improved by TiO2-RGO nanocomposites are potentially used for concrete enhancement, in which the epoxy is a key material for such action. In general, the improved epoxy and TiO2-RGO nanocomposites can be used in where the RGO use.

Reviewer 3 Report

An interesting work on “Enhanced Tensile Strength of Monolithic Epoxy with Highly Dispersed TiO2-Graphene Nanocomposites” has been presented. Manuscript is well explained. Following comments would be helpful to improve the paper.

  • Check for typos particularly for units (e.g. Page 6, Line 162-163 and Page 8, Line 233-273: cm-1 and cm-1).
  • Part of equation 2 is not fully visible.
  • Check alignment of captions for figures and tables.
  • Relabel figure 4 with a text of better visibility/font size.
  • Caption missing for figure 8, please add.
  • References from recent years are missing and should be included.

Author Response

An interesting work on “Enhanced Tensile Strength of Monolithic Epoxy with Highly Dispersed TiO2-Graphene Nanocomposites” has been presented. Manuscript is well explained. Following comments would be helpful to improve the paper.

  • Check for typos particularly for units (e.g. Page 6, Line 162-163 and Page 8, Line 233-273: cm-1 and cm-1).

Response: Thanks for your examination. Corrected already.

  • Part of equation 2 is not fully visible.

Response: Thanks for your examination. Corrected already.

  • Check alignment of captions for figures and tables.

Response: Thanks for your examination. The caption for Figure 8 is missing. Corrected already.

  • Relabel figure 4 with a text of better visibility/font size.

Response: Thanks for your examination. Corrected already.

  • Caption missing for figure 8, please add.

Response: Thanks for your examination. Corrected already.

  • References from recent years are missing and should be included.

Response: Thanks for your examination. The references from recent years are added.